# Evaluating the Impact of Green Coffee Bean Powder on the Quality of Whole Wheat Bread: A Comprehensive Analysis

**DOI:** 10.3390/foods13172705

**Published:** 2024-08-27

**Authors:** Raima Das, Debmalya Banerjee, Deblu Sahu, Juwairiya Tanveer, Soumik Banerjee, Maciej Jarzębski, Sivaraman Jayaraman, Yang Deng, Hayeong Kim, Kunal Pal

**Affiliations:** 1Department of Biotechnology, Brainware University, Barasat 700125, Kolkata, India; raimsmoonsoon@gmail.com (R.D.); soumikban@gmail.com (S.B.); 2Department of Biotechnology and Medical Engineering, National Institute of Technology Rourkela, Rourkela 769008, Odisha, India; debmalyab60@gmail.com (D.B.); deblusahu.nit@gmail.com (D.S.); tanveerj0799@gmail.com (J.T.); mountshiva@gmail.com (S.J.); 3Powell Laboratories Pvt. Ltd., Salt lake City 700091, Kolkata, India; 4Department of Physics and Biophysics, Faculty of Food Science and Nutrition, Poznan University of Life Sciences, 60-637 Poznan, Poland; maciej.jarzebski@o2.pl; 5College of Food Science and Engineering, Qingdao Agricultural University, Qingdao 266109, China; dengyang719@hotmail.com; 6Institute of Food Industrialization, Institutes of Green Bioscience & Technology, Center for Food and Bioconvergence, Seoul National University, Daehwa-myeon, Pyeongchang-gun 25354, Gangwon-do, Republic of Korea; hara2910@snu.ac.kr

**Keywords:** green coffee bean powder, bread, palatability, polysaccharides, shelf life

## Abstract

The current investigation focuses on the effect of different concentrations of green coffee bean powder (GCBp) on the physicochemical, microbiological, and sensory characteristics of whole wheat bread (WWB). C1 bread formulation (containing 1% GCBp) exhibited the highest loaf volume, suggesting optimal fermentation. Moisture analysis revealed minor alterations in the moisture retention attributes of the bread formulations. Impedance analysis suggested that C1 exhibited the highest impedance with a high degree of material homogeneity. Swelling studies suggested similar swelling properties, except C5 (containing 5% GCBp), which showed the lowest swelling percentage. Furthermore, color and microcolor analysis revealed the highest L* and WI in C1. Conversely, higher concentrations of GCBp reduced the color attributes in other GCBp-containing formulations. FTIR study demonstrated an improved intermolecular interaction in C1 and C2 (containing 2% GCBp) among all. No significant variation in the overall textural parameters was observed in GCBp-introduced formulations, except C2, which showed an improved gumminess. Moreover, the TPC (total phenolic content) and microbial analysis revealed enhanced antioxidant and antimicrobial properties in GCBp-incorporated formulations compared to Control (C0, without GCBp). The sensory evaluation showed an enhanced appearance and aroma in C1 compared to others. In short, C1 showed better physicochemical, biological, and sensory properties than the other formulations.

## 1. Introduction

Bread has been accepted as one of the oldest processed foods in history [1]. It is a well-known bakery product consumed worldwide; however, its consumption is predominant in Europe, Africa, and some parts of the Middle East [2,3]. The vital ingredients involved in breadmaking include flour, water, and yeast. Moreover, to enrich the texture, aroma, flavor, and taste of bread, some additional components such as milk, oil, sugar, eggs, and salt are often added to the dough [4]. Around the world, there are countless varieties of bread, each with unique qualities. These include bread prepared using a variety of grains, ingredients, and production methods, allowing them to be made with a wide range of properties, including shapes, sizes, flavors, texture, taste, etc. [5]. Traditionally, wheat flour is the primary constituent in bread production. Its composition comprises starch, gluten, non-starch polysaccharides (e.g., arabinoxylans, β-glucan, cellulose, arabinogalactan-peptides, etc.), and lipids (particularly triglycerides) in small quantities [6]. In recent years, refined flour generated from the milling process of whole wheat grain has been extensively used in breadmaking by the food processing industries. Though the use of refined flour improves the exquisite value of white bread, there is a significant reduction in the fiber and nutrient content of white bread [7]. In this regard, recent investigations have observed some detrimental health effects, such as type 2 diabetes mellitus (T2DM), posed by the regular consumption of refined flour. Refined flour has also been associated with other chronic diseases (e.g., inflammatory bowel disease, non-alcoholic fatty liver disease, etc.) attributed to the low dietary fiber content and high glycemic index of refined flour [8]. Thus, the perfect bread should have a lower glycemic index and be a significant source of protein, antioxidants, vitamins, magnesium, and trace minerals to overcome this problem [9].

Although industries still prefer refined flour as a standard resource in breadmaking technology, consumer demand for whole wheat bread has increased significantly in recent years [10]. This shift is partly due to a few epidemiological studies highlighting the health benefits of consuming whole grains and grain-based food formulations. These benefits include a lower risk of obesity, type 2 diabetes, carcinogenesis, and cardiovascular disease linked to oxidative stress [11]. The health-conferring benefits of whole wheat flour derive from its nutritional value, as it comprises bran and germ layers of the wheat kernel, which are rich sources of several ingredients, including essential amino acids (e.g., glutamic acid, tryptophan, etc.), dietary fibers (e.g., cellulose, hemicelluloses, etc.), minerals (e.g., Fe, Cu, Zn, Mg, Ca, etc.), fat-soluble vitamins (e.g., Vit K and E) and some unsaturated fatty acids (e.g., triglycerides). In contrast, white or refined flours retain only the endosperm [12]. Further, numerous phytochemicals, mainly alkylresorcinols (AR) and phenolic compounds, are also present in wheat grain, which have been shown to have anti-inflammatory and antioxidant properties. Hence, whole wheat bread and bakery products are widely accepted as an ideal source of nutrition and energy [13,14].

Unfortunately, the unsaturated fatty acids in WWF are susceptible to oxidation when exposed to high temperatures of baking conditions, which reduces the overall antioxidant potential and lifespan of the end product [15]. Thus, the current food production strategies include preserving food components and introducing bioactive components such as natural antioxidants in food to achieve pro-health attributes [16]. Recently, green coffee bean (GCB) powder has been incorporated into several food ingredients (e.g., chocolates, energy bars, cakes, cookies, capsules, beverages, etc.) due to its unique nutritional properties and composition [17]. Investigations have shown that GCB is a naturally occurring raw ingredient rich in phenolic antioxidants. In addition, unroasted coffee beans (CBs) or GCBs are believed to encourage metabolism, which helps reduce body weight and prevent obesity, due to their abundance of bioactive phenolic acids [18]. Scientifically, it has been confirmed that these phenolic compounds, especially chlorogenic acids in GCBs, are primarily responsible for antimutagenic, anticarcinogenic, and antioxidant activities [19]. As a result, consumption of GCB-fortified food products can contribute to health benefits [20]. Since GCBs contain a high amount of chlorogenic acid compared to roasted CBs, they exert various health benefits, especially antioxidant activity and antimicrobial properties [21]. Dziki et al. (2016) incorporated GCB powder (GCBp) (*Coffea arabica*) in whole wheat flour bread, with the concentration varying between 1 and 5% (*w*/*w*). Authors have reported an improved antiradical activity and overall consumer acceptability of the bread formulations enriched with GCB powder. Still, their study did not perform an in-depth analysis of these breads by analytical methods like impedance analysis, swelling percentage studies, FTIR spectroscopy, microbial count analysis, and sensory evaluation [22]. Although adding functional ingredients to bread is common, the effect of raw GCBp in whole wheat flour bread (WWB) fortification has little evidence. Accordingly, in this study, the effect of incorporating GCBp at varying concentrations (0–5% *w*/*w*) in WWB formulations was investigated. To thoroughly investigate various physicochemical attributes, several characterization studies, such as moisture analysis, swelling percentage analysis, FTIR spectroscopy, color and microcolor analysis, and evaluation of electrical (impedance spectroscopy) and biological (antimicrobial) properties, were conducted. 

## 2. Materials and Methods

### 2.1. Materials 

#### 2.1.1. Breadmaking Materials

Whole wheat flour (Classic Wheat Flour Atta, Avent Agro Pvt. Ltd., Delhi, India) was procured from Flipkart Grocery. The composition of the WWF, as provided by the manufacturer, comprised 10.36% protein, 77.16% carbohydrates, 10.47% dietary fiber, 1.43% fat, and 1.19% sugar. GCB powder (Carbamide Forte, HNCO Organics Pvt. Ltd., Gujarat, India), instant dry yeast (Goodrich Carbohydrates Ltd., Haryana, India), rice bran oil (Fortune rice bran health, Adani Wilmar Ltd., Gujarat, India), and commercial salt (TATA Salt, Tata Chemicals Ltd., Gujarat, India) and sugar (sulphur-free granular sugar, Uttam Sugar Mills Ltd., Uttarakhand, India) were purchased from the local market. The proximate composition of GCB powder was 12.02% (*w*/*w*) moisture, 2.66% (*w*/*w*) ash, 12.47% (*w*/*w*) crude protein, 64.75% (*w*/*w*) carbohydrates, 5.7% (*w*/*w*) crude fat, 87.98% (*w*/*w*) total solids, and 97.34% (*w*/*w*) volatile solids as determined by standard analytical and biochemical methods [23,24]. 

#### 2.1.2. Chemicals and Reagents

Gallic acid (Merck Specialities Pvt. Ltd., Mumbai, India), sodium carbonate (HiMedia Laboratories Pvt. Ltd., Maharashtra, India), ethanol (HiMedia Laboratories Pvt. Ltd., Maharashtra, India), and Folin–Ciocalteu reagent (Sisco Research Laboratories Pvt. Ltd., Mumbai, India) were used to estimate total phenolic content in formulations. Nutrient agar (HiMedia Laboratories Pvt. Ltd., Maharashtra, India) and sodium chloride (Loba Chemie Pvt Ltd., Mumbai, India) were utilized for microbial analysis.

### 2.2. Methods

#### 2.2.1. Formulation of WWBs (Whole Wheat Breads)

WWB formulations were prepared to contain GCBp at varying concentrations (0–5% *w*/*w*). The formulations were named C0 (Control; 0% *w*/*w* of GCBp), C1 (1% *w*/*w* of GCBp), C2 (2% *w*/*w* of GCBp), C3 (3% *w*/*w* of GCBp), C4 (4% *w*/*w* of GCBp), and C5 (5% *w*/*w* of GCBp). Then, the listed components (Table 1) were put into the breadmaking machine (Model:16010; KENT Atta and Bread Maker, KENT RO Systems Ltd., Noida, India). The machine was operated under the “Whole Wheat Bread” mode for kneading with intermittent resting, which continued for 70 min. Subsequently, the dough was fermented in the breadmaking machine for 120 min. Thereafter, the fermented dough was transferred to a baking pan (L:25 cm, B:13 cm, H:5 cm) and subjected to baking inside a microwave oven (Model: MC32j7035CT; Samsung Smart Oven, Samsung Electronics Pvt. Ltd., Kuala Lumpur, Malaysia) preheated to 180 °C for a duration of 5 min. Finally, baking was performed for a time period of 36 min, followed by cooling of the formulations for 1 h at 25 °C. 

#### 2.2.2. Measurement of Physical Dimensions 

The physical dimensions (length, breadth, and height) of the bread loaves were recorded at multiple points for each formulation using a digital vernier caliper (Model: IP54 metal case digital caliper; Advance Precision Measurement Est., An Nabiyah, Saudi Arabia). Thereafter, the average length (L_av_), average breadth (B_av_), and average height (H_av_) of each formulation were calculated. This step helped to account for the variations in the dimensions of the bread loaves. Subsequently, the average volume (V_av_) was computed using L_av_, B_av_, and H_av_ of each formulation employing Equation (1). V_av_ for each formulation was calculated and represented as mean ± standard deviation.
(1)$$V_{av}=L_{av}\times B_{av}\times H_{av}$$
where V_av_ denotes the average volume, L_av_ denotes the average length, B_av_ denotes the average breadth, and H_av_ denotes the average height. 

#### 2.2.3. Moisture of Crumb

A moisture analyzer equipped with a halogen heating source was employed in order to determine the moisture proportion of the formulations (Model: PGB1MB analyzer; Wensar Pvt. Ltd., Chennai, India). Consequently, the formulations were sized appropriately to ensure proper moisture measurement. Then, the formulations (~2 g) were kept in the aluminum pan inside the machine after cleaning the pan with a tissue dipped in ethanol, and their initial weight was noted down. The temperature was set at 180 °C in the moisture analyzer machine, and weight loss of the formulations was observed as the moisture loss. Finally, the moisture content (%) for each formulation was calculated using Equation (2) [21].
(2)$$Moisture\;content\left(\%\right)=\frac{A_{F}-A_{I}}{A_{F}}\times 100$$
where A_I_ denotes the initial weight of bread formulations and A_f_ denotes the weight culminating after the analysis. 

#### 2.2.4. Impedance Analysis

The impedance profiles of the formulations were assessed using impedance spectroscopy (Model: Impedance Breadboard Breakout for Analog Discovery 2, Digilent, National Instrument, Austin, TX, USA). The crumb portion was considered for the following study. Formulations were cut into pieces with the measurements of 4 cm (L) × 3 cm (B) × 4 cm (H). Then, the analyzer, containing a set of circular metallic probes (10 mm diameter), was introduced within the breadcrumbs. The impedance recordings were carried out in the frequency range of 1 Hz to 1 KHz.

#### 2.2.5. Study of Swelling Percentage

Formulations were initially cut into small cubes (2 cm × 2 cm × 2 cm), each nearly equal in weight. Subsequently, the bread cubes were kept in 100 mL beakers filled with 25 mL of water. Then, the beakers were placed inside a water bath (Model: Lmwb-8, Labman Scientific Instruments Pvt. Ltd., Chennai, India), maintained at a temperature of 37 °C. The final weight of each formulation was recorded at regular time intervals of 15 min for a period of 2 h followed by an interval of 1 h to a maximum duration of 5 h to determine the swelling percentage as per Equation (3). The experiment was performed in triplicates.
(3)$$SP\left(\%\right)=\frac{W_{F}-W_{I}}{W_{I}}\times 100$$
where SP (%) is the swelling percentage, W_I_ refers to the initial weight of the WWBs formulations, and W_F_ is the final weight (recorded after the swelling process).

#### 2.2.6. Colorimetry and Microcolorimetry

The color parameters $L^{*}$ (lightness), $a^{*}$ (color shift from red to blue), and $b^{*\;}$ (color shift from yellow to green) for the formulations were determined as per the method described in the previous study [25]. Then, WI (whiteness index), YI (yellowness index), and BI (brownness index) of the formulations were estimated from the $L^{*\;}$, $a^{*\;}$, and $b^{*\;}$. Equations employed for the calculation of WI (Equation (4)), YI (Equation (5)), and BI (Equation (6)) values are provided below [26].
(4)$$WI=100-\sqrt{{\left(100-L^{*}\right)}^{2}+{\left(a^{*}\right)}^{2}+{\left(b^{*}\right)}^{2}}\;$$
(5)$$YI=142.86\times \frac{b^{*}}{L^{*}}\;$$
(6)$$BI=\frac{\left\{100\left(X-0.31\right)\right\}}{0.17}\;$$
where $X=\frac{\left(a^{*}+{1.75L}^{*}\right)}{\left(5.645XL^{*}+a^{*}-3.01{2b}^{*}\right)}$

The parameters WI, YI, and BI allude to the whiteness index, yellowness index, and brownness index, respectively.

In addition to colorimetry, the color parameters of WWB formulations were also evaluated at the microscopic level by determining the $L_{m}^{*}$, $a_{m}^{*}$, and $b_{m}^{*}$ values for each formulation. Whiteness index (${WI}_{m}$), yellowness index ${(YI}_{m})$, and brownness index ${(BI}_{m})$ were computed using the values of $L_{m}^{*}$, $a_{m}^{*}$, and $b_{m}^{*}$ employing Equation (4), Equation (5) and Equation (6), respectively. The imaging was done using a handheld digital microscope (Model: 44308; Celestron Pro Microscope, Celestron, CA, USA).

#### 2.2.7. FTIR Study

The infrared absorption spectral profiles of the formulations were determined using an FTIR spectrophotometer (Model: Alpha-E; Bruker, Bremen, Germany) equipped with an attenuated total reflectance (ATR) module comprising a zinc selenide (ZnSe) crystal. The spectra of all the WWB formulations were recorded in the 4000–500 cm^−1^ region, with 25 scans at a spectral resolution of 4 cm^−1^ performed on each formulation [27]. 

#### 2.2.8. Texture 

The texture profile analysis (TPA) of all the WWBs was performed to determine the textural properties of all the bread formulations using a texture analyzer (Model: Texture Analyzer HD Plus; Stable Micro Systems, Godalming, UK). In this investigation, the crumb portions of the formulations (dimensions: 20 mm × 20 mm × 20 mm) were compressed using a flat probe (diameter: 30 mm). Two compression cycles were performed at a speed of 1 mm/s and a 50% deformation rate by maintaining a 5 s break between each cycle. The major textural parameters, such as hardness, springiness, cohesiveness, chewiness, resilience, and gumminess, were obtained from this study. Herein, the experiments were conducted in triplicates [28].

#### 2.2.9. Estimation of Total Phenolic Content

##### Formulation Extract Preparation

The ethanolic extracts of the respective formulations were employed to retrieve information about the total phenolic content. For this study, the crumb portion was considered. Here, the crumb was taken and minced into small pieces. Subsequently, 1 g of the minced formulations was weighed and immersed in 9 mL of 80% ethanol. Thereafter, the mixtures were ultrasonicated for 15 min at 25 °C for homogenization, followed by centrifugation (Model: Spinwin MC-01 microcentrifuge; Tarsons Products Ltd., Kolkata, West Bengal, India) at 25 °C in 8944× *g* for 15 min. The filtrates were obtained using filter paper (diameter: 125 mm) and stored at 4 °C for further analysis.

##### Total Phenolic Content

The total phenolic content of the WWB formulations was estimated according to the Folin–Ciocalteu (FC) method as per the method described by Singleton et al. (1999) [29]. In summary, a 0.5 mL extract from each bread formulation was dissolved separately in distilled water of volume 1.8 mL, followed by the addition of 0.2 mL of 10% FC reagent. The formulations were incubated for 5 min. Subsequently, 2 mL of 7% sodium carbonate and 0.8 mL of distilled water were incorporated and mixed uniformly. Furthermore, the formulations were incubated for a period of 90 min at 25 °C, and the absorbance was recorded at 750 nm wavelength using a UV–VIS spectrophotometer (Model: UV 3200 Double Beam Spectrophotometer, Labindia Pvt. Ltd., Thane, India). The gallic acid solution was employed as a standard for this study. The results are represented as mg GAE/100 g of dry mass.

#### 2.2.10. Microbial Analysis of Bread

##### Assessment of Total Viable Count (TVC)

Microbiological evaluation of all the formulations was carried out using the viable count method to compare the antimicrobial properties of the formulations by using nutrient agar (2% NA) as the growth medium. Initially, 1 g of the formulation was dissolved in 9 mL of saline solution (0.85% NaCl) to prepare the stock. The stock solutions were diluted 3 times (10^−1^ to 10^−3^) using a sterile saline solution. Subsequently, ~25 mL NA media was poured into sterilized Petri plates (8.5 cm diameter) and solidified for 1 h. Thereafter, 0.1 mL of formulation from the 10^−3^ dilution was taken and spread evenly over the growth media using a sterile spreader. At last, these plates were incubated overnight (12 h) at 37 °C. The experiments were conducted for 3 days, from the 0th day to 2nd day, and the TVC of the bacterial colonies was observed at 24 h intervals. Finally, the total plate count of the bacterial colonies is represented as colony-forming units (CFU) per gram of WWB formulations.

#### 2.2.11. Sensory Analysis

The sensory evaluation of the formulations involved 20 semi-trained panelists, comprising individuals aged between 20 and 35. For this study, a single blindfolded evaluation was performed. At first, the WWBs were whittled, and the panelists assessed each slice. Using a scale of 1 to 5, the formulations were rated according to their perception levels, where 1 indicated extreme dislike, 2 expressed dislike, 3 indicated neither like nor dislike, 4 represented like, and 5 represented extreme like. The sensory attributes evaluated in this study included appearance, color, aroma, softness, and overall impression. 

#### 2.2.12. Statistical Analysis

The studies were conducted in triplicates, and the ramifications are denoted as the mean ± standard deviation. SPSS software was employed in order to check the statistical parameters (version 20, IBM Inc., Chicago, IL, USA). Significant differences between the values were determined using ANOVA (one-way analysis of variance), which was followed by Tukey post hoc test. A significance level of *p* < 0.05 was utilized to identify meaningful disparities among the outcomes.

## 3. Results

### 3.1. Physical Evaluation of WWB Preparations

The visual appearance of the control bread formulation (C0) showed a yellowish-brown color. This might be attributed to the presence of carotenoids, especially the lutein pigment in the WWF [30]. Thereafter, the addition of GCBp in C1 to C5 resulted in the development of a brownish shade in the prepared formulations (Figure 1), which can result from the Maillard reaction (MR). MR occurs among the various components (e.g., polyphenols and amino acids) present in GCBp with the reducing sugar in the bread dough [31]. During the high-temperature baking process, amino acids react with sugar molecules to form an amino-acid–sugar complex, initiating the MR, which contributes to odor development and imparts food browning [32,33]. Overall, an increment in the GCBp resulted in the formation of formulations with darker shades. This can be explained by the formation of an increased quantity of amino-acid–sugar complexes due to MR. Similar observations have also been reported in [31].

Further, formulations were vertically cut to observe the internal structure and color of the crumb (Figure 2). It was observed that C0 exhibited a uniform distribution of small pores with few large pores. After incorporating 1% GCBp in C1, significantly larger pores were observed, along with a homogeneous distribution of small pores compared to Control (C0). However, further increasing the GCBp content reduced the pore size in C2. Thereafter, subsequent enhancement in the GCBp content in C3 led to the development of larger and uniform pores. It was observed that the homogeneity of large pores was maintained in C4 and C5, although the number was comparatively less than in C3 (Figure 2). Generally, the porosity of WWB bread is correlated with improved yeast activity and fermentation process, due to which CO_2_ gas bubbles are trapped within the bread crumb [34]. The observations noted in C1, C2, C4, and C5 suggest the improvement and uniformity in the porosity of the formulations, which could be attributed to the increase in bread volume [35]. This might be correlated to the enhanced yeast fermentation processes and activity within the dough compared to C0 upon the addition of GCBp [36].

Evaluation of the quality and desirability of food products is usually carried out by physical examination. The physical dimensions were measured, and the average volume (V_av_) was calculated as described in Section 2.2.2. It was found that the L_av_ of formulation C0 was 172.21 ± 1.27 mm. After adding GCBp, C1 showed the highest L_av_ (192.83 ± 0.55 mm; *p* < 0.05) of the bread loaf compared to all the other formulations. A further increment in the GCBp content in C2 led to a significant decrease in L_av_ compared to the formulations C0 and C1 (*p* < 0.05). C2 showed the lowest L_av_ among all the formulations (*p* < 0.05). Thereafter, a further replacement of WWF with 3%, 4%, and 5% GCBp (*w*/*w*) in C3, C4, and C5 contributed to a rise in the L_av_ compared to C2 and C0 (*p* < 0.05), while the L_av_ was smaller than in C1 (*p* < 0.05) (Figure 3A). 

The B_av_ of C0 was 72.08 ± 0.31 mm. After the addition of 1% (*w*/*w*) GCBp in bread, the B_av_ of C1 had increased to 86.89 ± 0.24 mm (*p* < 0.05), which was the highest among all the formulations. A subsequent increase in GCBp in C2 had notably reduced the B_av_ (67.23 ± 0.24 mm, *p* < 0.05). In contrast, when the content of GCBp was further increased, it was observed that the B_av_ had enhanced in C3 (74.15 ± 0.63 mm), C4 (77.62 ± 0.52 mm), and C5 (78.90 ± 0.30 mm) compared to C2 and C0 (*p* < 0.05), as illustrated in Figure 3B. The observed increment in the B_av_ and L_av_ of C1 could be attributed to the even distribution of pores with an augmentation of porosity. However, C2 exhibited the lowest L_av_ and B_av_ parameters compared to all other formulations. This observation might be attributed to decreased porosity and non-uniform distribution of pores in the crumb structure (Figure 2). Moreover, C3, C4, and C5 demonstrated relatively higher L_av_ and B_av_ values than C2. This could be correlated to the enhanced porous structure of the crumb after incorporation of GCBp.

The H_av_ of C0 was 49.80 ± 0.02 mm. Subsequently, it was observed that there was a reduction in H_av_ in C1 (47.08 ± 0.22 mm, *p* < 0.05) and C2 (40.42 ± 0.03 mm, *p* < 0.05) (Figure 3C). Further addition of GCBp led to an enhancement in H_av_ in C3 (46.06 ± 0.33 mm, *p* < 0.05). However, it was noted that upon an increase in the GCBp amount in WWB, the H_av_ of C4 (43.61 ± 0.32 mm) and C5 (42.42 ± 0.06 mm) was significantly reduced as compared to C3 (*p* < 0.05) (Figure 3C). The overall elevation in H_av_ in C1, C3, C4, and C5 might be correlated to the porous structures of these bread formulations, which were noted in the cross-section view (Figure 2), which suggests that the addition of GCB powder promoted the fermentation process in dough by increasing CO_2_ production which led to larger bread size. However, C2 exhibited the lowest height among all the WWB formulations. This might be attributed to the fact that the incorporation of GCBp could interfere with the pore formation in C2 [37].

The V_av_ of C0 was 6.18 × 10^5^ ± 0.03 × 10^5^ mm^3^. In C1, there was an increase in the V_av_ (7.88 × 10^5^ ± 0.04 × 10^5^ mm^3^, *p* < 0.05) compared to C0. In fact, C1 exhibited the highest V_av_ when compared with other formulations. Furthermore, an increase in the amount of GCBp caused a significant decrease in the bread volume of C2 to 4.4 × 10^5^ ± 0.08 × 10^5^ mm^3^ (*p* < 0.05), which was the lowest among all the formulations. Thereafter, there was an increase in the V_av_ of C3 (5.99 × 10^5^ ± 0.07 × 10^5^ mm^3^), C4 (6.04 × 10^5^ ± 0.18 × 10^5^ mm^3^), and C5 (6.09 × 10^5^ ± 0.07 × 10^5^ mm^3^), compared to C2 (*p* < 0.05). However, there was no significant difference among the V_av_ of C3, C4, and C5 (*p* > 0.05) (Figure 3D). In summary, it can be concluded that GCBp content in the lowest amount increased the loaf volume of C1 as compared to C0. The loaf volume of other GCBp-containing bread formulations (C2, C3, C4, and C5) was decreased compared to C1. A similar observation was reported by Onacik-Gur et al. (2021), where the authors reported an increase in the loaf volume when natural plant additives (e.g., tea leaf extract) were added in the lowest concentrations. This was attributed to the optimal fermentation level. Hence, it can be estimated that adding GCBp at the lowest concentration (1%) helped to achieve optimal dough fermentation, thereby resulting in a higher loaf volume than the other formulations [38]. Further, the overall variation in the loaf volume was markedly different from the study published by Dziki et al. (2016) [22], where it was found that the addition of GCBp in lower proportions (1% and 2%) did not alter the loaf volume, whereas, in higher proportions of GCBp (3%, 4%, and 5%), there was a significant increase in the loaf volume of the WWB. This observation can be explained by the change in the fermentation conditions. The dough was fermented in a fermentation chamber (ICH 256, Memmetr, Germany, Dusseldorf) by Dziki et al. (2016) [22]. In contrast, in our study, the fermentation was carried out in a breadmaking machine (Model:16010; KENT Atta and Bread Maker, KENT RO Systems Ltd., Noida, India). Another plausible reason could be the change in the composition. Dziki et al. (2016) [22] did not include sugar in dough preparation. However, the dough we prepared consisted of sugar in a definite quantity as per the method reported by our group in a previous study [39]. Hence, it was expected that the difference in the baking conditions and composition could show deviation from the results reported by Dziki et al. (2016) [22]. These observations can be correlated with the findings of Ibrahim et al. [40].

### 3.2. Moisture Content

The moisture content of the bread formulations significantly influences the bread’s sensory attributes, palatability, and overall acceptability [41]. Several studies have shown that after baking, bread undergoes some changes during storage, which include physical and chemical processes like crumb firming, water movement within the bread crust and crumb, flavor loss, and the risk of microbial spoilage [42]. Hence, the analysis of the bread moisture profile is an absolute necessity. Moisture content analysis revealed that the moisture content of C0 was 51.53 ± 0.05%. Upon increase in the GCBp concentration, it was observed that the moisture content of C1 was similarly valued with that of C0 (*p* > 0.05) (Figure 4). A further enhancement of the GCBp content resulted in a slight increase in the moisture content of C2 (52.20 ± 0.04%) and C3 (52.23 ± 0.01%), which was statistically significant compared to all other formulations (*p* < 0.05). C4 (51.53 ± 0.1%) and C5 (51.57 ± 0.05%) exhibited a slight reduction in the moisture content compared to C2 and C3. The moisture content of C4 was similar to that of C0 and C1 (*p* > 0.05), while that of C5 was similar to that of C0 (*p* > 0.05). Among C4 and C5, the moisture content was similar (*p* > 0.05). However, there was a slight variation in the moisture content, as apprehended from the observations. This can be attributed to the saturation point of water absorption by the polysaccharides, viz. cellulose and hemicellulose present in GCBp, beyond which further incorporation of GCBp will not cause a corresponding increase in the moisture retention capability, as observed from the results [43,44]. Moreover, the inclusion of GCBp does not significantly affect the enzymatic action of yeast, which breaks down the starch molecules into simple sugars. This can explain the similar hydration attributes in the formulations [45].

### 3.3. Analysis of Impedance

The electrical attributes of the materials are analyzed using a technique known as electrical impedance spectroscopy (EIS). This involves noting the alterations in the impedance profiles as the frequency of a particular voltage is changed [46]. Figure 5 represents the Bode plots, which demonstrate the impedance and phase responses of the formulations. The observations showed that C1 exhibited relatively higher impedance among all the formulations at lower frequencies ranging from 1 Hz to 100 Hz. This result can be correlated with the highest volume of C1, which confirmed that the gas bubble entrapment in C1 was the highest (Figure 5). However, the overall impedance profile did not reflect any significant difference between the formulations throughout the frequencies. Moreover, it was observed that at lower frequencies, the phase response was −70°, but as the frequency was gradually enhanced, the phase response shifted to −10° for all the formulations. This might be attributed to the capacitive behavior of the formulations [47,48]. Further, the impedance profile was fitted to an (RQ)Q model (Equation (7)), which is one of the simplest electrical models to elucidate the electrical attributes of soft materials [49]. The parallel resistance and the constant phase element (CPE) represent the bulk properties of the materials, whereas the electrode and formulation interfaces are represented as a CPE in series with the above parallel equivalent circuit.
(7)$$Z_{eq}=\frac{R}{1+{\left(j\omega \right)}^{n}\times QR}+\frac{1}{1+{\left(j\omega \right)}^{n_{1}}\times Q_{1}}$$
where Z_eq_ denotes impedance, R denotes resistance associated with the bulk property of the formulations, Q denotes constant phase element associated with the bulk property of the formulations, Q_1_ denotes constant phase element associated with the formulation–electrode interfaces, n denotes the homogeneity constant associated with Q, and n_1_ denotes the homogeneity constant associated with Q_1_. 

Various electrical parameters were determined from the modeling of the experimental data to the (RQ) Q electrical model (Table 2). The fitting of the experimental data with the electrical model suggested an excellent fit, with a correlation of >0.99. It can be observed that the CPE (Q) of C1 was the lowest. This can be corroborated with the highest volume of C1, as stated in previous studies. Additionally, C1 showed the highest *n* value amongst all the formulations, suggesting better compositional homogeneity of C1 over the others, which is evident from the physical examination studies. 

### 3.4. Swelling Percentage

The analysis of the swelling percentage is essential to determine the quality of bread, as it reflects its moisture-holding capacity. This feature directly impacts the bread quality, texture, and crumb structure [50]. The study was carried out for a duration of 5 h. It was found that C0 exhibited the highest swelling percentage (SP) profile among all the formulations throughout the experiment (Figure 6). The SP of the formulations, namely, C1, C2, C3, and C4, was considerably lower compared to C0, but their differences were meager. C5 showed the lowest SP. Interestingly, the SP of C0 was saturated after 180 min, while the other formulations showed an early saturation (~150 min). 

After that, the SP of the prepared formulations at the end of 5 h was analyzed statistically (Figure 6). It was observed that C0 had an SP of 148.40 ± 10.38%. Overall, the addition of GCBp led to the reduction of the SP. However, the SI of every formulation at the end of 5 h, except C5, was statistically insignificant (*p* > 0.05). The reduction in the SP of bread when the concentration of GCBp is increased could be linked to the presence of compounds, such as fatty acids and proteins, in GCBp. According to previous researchers, these compounds tend to inhibit the starch gelatinization in the dough, resulting in a decreased swelling power of the bread [51].

### 3.5. Colorimetry and Microcolorimetry

Colorimetric analysis is essential for evaluating the sensory attributes of food items, particularly to characterize their color. The color attributes, namely, $L^{*\;}$, $a^{*\;}$, and $b^{*\;}$, were utilized to analyze the color of the formulations according to the CIELAB system [52]. $L^{*\;}$ denotes lightness, which reflects variations in grayscale values. It ranges from 0 to 100; this correlates with the color scale, which reflects black and white colors. The values of the parameters $a^{*\;}$ and $b^{*\;}$, however, lie between −120° and +120°. The value of $a^{*\;}$ towards the positive axis signifies redness, whereas the $a^{*\;}$ value tending towards the negative axis suggests a green hue. Similarly, the negative $b^{*\;}$ axis indicates a tendency toward blueness, and the positive $b^{*\;}$ axis implies a yellowish tint [53]. The $L^{*\;}$ value of C0 was recorded to be 61.73 ± 1.07. Thereafter, on incorporating GCBp in bread, a rise in $L^{*\;}$ value was observed in C1 (72.34 ± 1.4, *p* < 0.05), which was significantly highest among the formulations (Table 3). However, further addition of GCBp caused a significant reduction in the $L^{*\;}$ value of C2, C3, C4, and C5. The $L^{*\;}$ values of C2, C3, and C4 were similar among themselves and also with C0 (*p* > 0.05). Notably, C5 showed the lowest $L^{*\;}$ value (58.11 ± 0.83), which was similar to that of C4 (59.77 ± 0.83, *p* > 0.05). Obtaining the lowest $L^{*\;}$ value in C5 could be attributed to the increase in the MR between GCBp and sugar molecules, resulting in a loss of whiteness in bread compared to other formulations [54]. This finding aligns with the earlier observations depicted in Figure 1, which revealed that the browning of the bread increased due to the elevated MR when GCBp was added in higher concentration. Additionally, the highest $L^{*\;}$ value in C1 suggests the retention of lightness in bread. 

The $a^{*\;}$ value of C0 was 6.55 ± 1.61. C1, C2, C3, and C5 showed statistically similar $a^{*\;}$ values, as compared to C0 (*p* > 0.05). C4 (10.52 ± 1.50, *p* < 0.05) exhibited comparatively higher $a^{*\;}$ values in relation to all other formulations (Table 3). It has been reported that a positive $a^{*\;}$ value indicates a reddish-brown hue in bread formulations due to caramelization or enzymatic reaction in bread dough [55]. This suggests that an elevated $a^{*\;}$ value might cause a reddish-brown tint in GCBp-containing bread formulations. The $b^{*\;}$ value of C0 was 44.91 ± 1.14. There was no statistical difference between the $b^{*\;}$ values of C0, C1 (41.68 ± 1.18), and C2 (44.97 ± 3.40) (*p* > 0.05). Interestingly, C3, C4, and C5 showed a significant enhancement in $b^{*}$ value, which was statistically elevated compared to C0, C1, and C2 (*p* < 0.05). However, C3, C4, and C5 showed similar b* values (*p* > 0.05) (Table 3). From the previous studies, it was observed that a positive $b^{*\;}$ value suggested a yellowish hue of the bread formulations due to the presence of carotenoid pigment in the bread dough [56]. In our study, a comparison of $a^{*\;}$ and $b^{*\;}$ values suggest a predominant yellow hue in the prepared formulations. 

The WI value of C0 was 40.60 ± 1.29. C1 exhibited a significant increase in the WI value (49.53 ± 0.31) compared to C0 (*p* < 0.05). Thereafter, C2 showed a notable reduction in WI value, as compared to C1 (*p* < 0.05). C3, C4, and C5 showed a reduction in WI on further incorporation of GCBp, as compared to C2 (*p* < 0.05). Nonetheless, C3, C4, and C5 showed insignificant variation in the WI values (*p* > 0.05) (Table 3). The YI of C0 was 103.98 ± 3.00. C1 exhibited a decrease in the YI compared to C0 (*p* < 0.05). Thereafter, C2 showed a higher YI value than C1 (*p* < 0.05), while there was no statistical difference between C0 and C2 (*p* > 0.05). Moreover, the YI values escalated in C3, C4, and C5, as compared to C2 (*p* < 0.05). Nevertheless, there was no statistical significance between C3, C4, and C5 (*p* > 0.05) (Table 3). Concerning YI values, the BI values followed a similar pattern. It was noted that the BI value of C0 was 124.56 ± 7.10. C1 displayed a significant reduction in BI (Table 3) compared to C0 (*p* < 0.05). Further addition of GCBp increased the BI of C2, compared to C1 (*p* < 0.05), but there was no marked variation between the values of C0 and C2 (*p* > 0.05). Additionally, C3, C4, and C5 showed an enhancement in BI values in accordance with other formulations (*p* < 0.05), while the BIs of C3, C4, and C5 were similarly valued (*p* > 0.05).

The colorimetric data suggest that C1 showed the least browning effect, with correspondingly higher $L^{*\;}$ and WI values. This can be explained by the previous investigations, which state that the introduction of phenolic compounds (GCBp) into the bread matrix at lower amounts stabilizes the gluten network, while a higher content of GCBp disrupts the gluten network [57]. Researchers have reported that the destabilization of the gluten network causes an enhancement in the MR, which leads to an increased browning in bread [58]. Accordingly, our finding suggests that as C1 consisted of the lowest concentration of chlorogenic acid (as the GCBp content was the lowest), the browning of C1 was minimal.

Further, microcolorimetry analysis was performed to understand the microscale variations in the color information. The findings showed that C1 exhibited the highest $L_{m}^{*}$ and ${WI}_{m}$ as compared to other formulations (Table 4). This suggests a low browning effect in C1, corroborating the findings observed from the colorimetric studies. Therefore, adding GCBp at lower concentrations (1% GCBp) improved the visual attributes of the formulations, as explained by the results obtained via colorimetric and microcolorimetric investigations.

### 3.6. FTIR Study

Fourier transform infrared spectroscopy analysis (FTIR) was performed to understand potential variations in the interactions amongst the key functional groups of the components of WWB. After analyzing the FTIR spectra of bread formulations, the major peaks were identified at 3309, 2971, 1743, 1647, 1452, 1383, 1148, 1014, and 681 cm^−1^ (Figure 7). The peaks corresponding to the wavenumber region of 4000–1200 cm^−1^ represent the functional groups, while the peaks falling under the range of 1200–500 cm^−1^ are considered the fingerprint of specific molecules [59,60]. A visual analysis of the FTIR spectra suggests that the FTIR profiles were similar. All the bread formulations displayed a broader peak at the wavenumber 3309 cm^−1^. This might correspond to the molecular interactions of the hydroxyl (–OH) moieties, especially stretching vibrations [61]. The peak intensities of C0, C1, and C2 displayed an overlapping pattern and appeared higher than those of C3, C4, and C5. A previous report by Obeidat et al. (2018) suggested that a lower concentration of GCBp in bread might enhance the overall intensity of –OH stretching vibrations due to the interaction between the phenolic compounds of GCBp with starch and protein in the bread matrix [62]. However, with higher concentrations of GCBp, the increased amount of phenolic compounds can lead to their self-association, which might reduce the intensity of the –OH stretching vibrations in the bread formulations. This relates to the fact that there is a direct correlation between our findings and the observations reported by Obeidat et al. (2018). Moreover, this observation can be corroborated by the findings obtained from the colorimetric analysis, as described in Section 3.5. It was found that a lower concentration of GCBp stabilizes the gluten network in the bread matrix, which aided in the enhancement of the –OH stretching peak intensities. On the other hand, a higher GCBp content might have led to the destabilization of the gluten network, which resulted in the decrement of the –OH stretching peak intensities [57].

On further analysis, it could be identified that at 2971 cm^−1^, a narrow but sharp peak could be observed in all the formulations. This can be particularly related to the stretching vibration of the C–H bonding in the alkyl moiety, thereby indicating the presence of lipid molecules [63]. One of the reasons might be the presence of fatty acids (e.g., palmitic acid and linoleic acid) in GCBp or the TAGs (triacylgycerol) of oil incorporated into the bread formulations [64]. Thereafter, narrow, sharp peaks were noted at the corresponding wavenumbers 1743 cm^−1^ and 1647 cm^−1^ throughout the formulations. This might be due to C=O stretching vibrations in amide groups that are present in gluten protein. Gluten is one of the primary protein constituents of WWF [65]. Furthermore, all the formulations displayed similar intensities at the wavenumber 1743 cm^−1^ (amide-I). However, there was a notable enhancement in the intensity of C=O stretching vibrations in C1 and C2 at 1647 cm^−1^ (associated with amide-I) as compared to C3, C4, and C5.

Moreover, at the wavenumber 1452 cm^−1^, a small peak could be observed, which can be corroborated by the C–H bending, as per previous reports [66]. On careful observation, it was found that formulations C1 and C2 displayed higher intensities at this wavenumber, owing to the presence of gluten protein in formulations. Thereafter, another sharp peak at the region 1383 cm^−1^ could be found in all the formulations. Kanazawa et al. (2021) reported that the peak at this wavenumber indicates the presence of C–N/N–H bending of lysine residue [67]. Our findings showed that C1 and C2 displayed a higher intensity at 1383 cm^−1^ than all other formulations. From this observation, it could be stated that a lower concentration of GCBp might have enhanced the overall availability of lysine (amino acid) in the gluten matrix, possibly leading to higher peak profiles [68].

There were notable, sharp peaks at the wavenumbers of 1148 cm^−1^ and 1014 cm^−1^ in all the formulations. The origin of the peaks could be traced to the stretching of C–O, which are present within the –CH_2_OH functional groups present in starch [69]. C1 and C2 correspondingly displayed a higher peak intensity at the wavenumber of 1148 cm^−1^ compared to other formulations. This can be corroborated by enhanced interaction and an interplay between the starch and phenolic compound when GCBp is added in lower amounts, thus amplifying the C–O bond vibrations in the bread matrix [62]. Conversely, higher concentrations of GCBp lead to greater phenolic self-association, reducing their interaction with the starch in the bread matrix. Thereafter, all the formulations exhibited similar intensity at 1014 cm^−1^. Subsequently, a small peak at 681 cm^−1^ was displayed by all the formulations, which indicates the C–O–C stretching of starch molecules [61]. Visually, all the formulations showed similar intensity for this peak. Overall, it could be observed from the FTIR spectra that the addition of GCBp in lower content had improved the inter-molecular interactions in C1 and C2 compared to others.

### 3.7. Texture of Bread

The texture profile analysis (TPA) was carried out in order to simulate the two-bite test, encompassing the initial and subsequent compression cycles [70]. This test plays a crucial role in ensuring the quality of the developed food products by evaluating attributes like cohesiveness, hardness, and others. Moreover, it assists in evaluating consumer perception of bread quality, which is a critical factor in food product development [71]. C0 showed a hardness of 645.75 ± 9.03 g, which was statistically similar to that of C1 (643.68 ± 2.43 g, *p* > 0.05). The hardness of C2 (637.22 ± 4.53 g) was similar to that of C0 (*p* > 0.05); however, it was significantly lower than that of C1 (*p* < 0.05). Like C1, the hardness of C3 (638.77 ± 14.97 g) and C4 (647.95 ± 6.53 g) were similarly valued as compared to that of C0 (*p* > 0.05) but significantly lower than that of C1 (*p* < 0.05). Also, C3 and C4 showed statistically similar hardness values to C2 (*p* < 0.05). The hardness value of C5 (643.77 ± 7.86 g) was the highest but had a similar hardness compared to C1 (*p* > 0.05) (Figure 8A). Overall, it could be observed that the hardness of all the bread formulations did not vary significantly upon the addition of GCBp.

Springiness is a crucial parameter in texture analysis, which elucidates the elastic properties of the crumb by determining the degree of recovery between the first and second compressions. C0 showed a springiness value of 0.9 ± 0.02. After that, upon incorporating lower amounts of GCBp, the springiness of C1 (0.88 ± 0.05) and C2 (0.89 ± 0.04) were similarly valued in comparison with C0 (*p* > 0.05). Furthermore, C3 (0.77 ± 0.10) showed a similar springiness as compared to C0, C1, and C2 (*p* > 0.05). Subsequently, the springiness of C4 (0.69 ± 0.08) was similarly valued to that of C3 (*p* > 0.05). C5 (0.81 ± 0.02) displayed a similar springiness in comparison with other formulations (*p* > 0.05) (Figure 8B). Overall, it was noticed that the addition of GCBp did not alter the springiness of the bread formulations compared to C0, except for C4.

The gumminess of C0 was 360.26 ± 12.30. Upon incorporation of GCBp, the gumminess of C1 (371.58 ± 59.25) was similarly valued compared to C0 (*p* > 0.05). Furthermore, C2 (388.73 ± 57.83) showed the highest gumminess among all the formulations (Figure 8C). However, the gumminess of the formulations C1 and C2 were similarly valued (*p* > 0.05). Thereafter, C3 (387.38 ± 38.35) and C4 (361.95 ± 36.77) exhibited a reduction in gumminess as compared to C2 (*p* < 0.05). However, there was no statistical significance between the gumminess of C3 and C4 and that of C0 (*p* > 0.05). Moreover, the gumminess of C5 (364.93 ± 41.36) was similarly valued to that of C0, C1, C3, and C4 (*p* > 0.05). Overall, all the GCBp-incorporated formulations exhibited similar gumminess to that of C0, except in the case of C2. 

The cohesiveness of C0 was 0.55 ± 0.02. After the addition of GCBp, C1 (0.62 ± 0.06) exhibited the highest cohesiveness. The cohesiveness of C2 (0.60 ± 0.05) was similar to that of both C0 and C1 (*p* > 0.05). C3 (0.47 ± 0.06) and C4 (0.45 ± 0.01) resulted in a monotonous decrease in the average cohesiveness (Figure 8D). However, their cohesiveness was similarly valued to the cohesiveness of C0 (*p* > 0.05). Also, the cohesiveness of C3 was similar to that of C2 (*p* > 0.05). Further, the cohesiveness of C5 (0.49 ± 0.05) was similar to that of all other formulations (*p* > 0.05). Overall, it could be observed that the variations in cohesiveness of the formulations were not much as compared to C0.

The chewiness of C0 was 324.41 ± 10.00. After adding GCBp, C1 (416.24 ± 42.39) showed a similar chewiness compared to C0 (*p* > 0.05). Thereafter, C2 (442.15 ± 69.49) and C3 (442.87 ± 55.25) exhibited the highest chewiness; however, they were similar to C1 (*p* > 0.05). In C4 (327.84 ± 16.75), the chewiness showed a significant reduction compared to C2 and C3 (*p* < 0.05). However, it was similarly valued to the chewiness of C0 and C1 (*p* > 0.05). Further, the chewiness of C5 (318.71 ± 13.83) was the lowest (*p* < 0.05) (Figure 8E). Among the GCBp-containing formulations, the chewiness was higher in formulations containing lower amounts of GCBp (C1, C2, and C3); however, at higher proportions of GCBp (C4 and C5), the chewiness was significantly decreased. 

The resilience of C0 was 0.27 ± 0.01. After the addition of GCBp, C1 (0.31 ± 0.05) and C2 (0.31 ± 0.04) showed similar resilience as compared to C0 (*p* > 0.05) (Figure 8F). Furthermore, C3 (0.20 ± 0.04) and C4 (0.17 ± 0.00) displayed a significant reduction in resilience when compared with the formulations C1 and C2 (*p* < 0.05). Thereafter, the resilience of C5 (0.22 ± 0.03) was similarly valued in relation to all other formulations (*p* > 0.05). Overall, it could be observed that the variation in resilience among the formulations was insignificant with respect to Control (C0). However, among the GCBp-containing formulations, C1 and C2 showed the highest resilience, while C3 and C4 showed the lowest resilience. 

Overall, it was observed that most of the texture parameters were similar in all the formulations (except gumminess and chewiness), as per the findings of Ibrahim et al. (2020) [40]. The gumminess of C2 was the highest. On the other hand, the gumminess in all other formulations was similar. Further, among the GCBp-incorporated formulations, C1, C2, and C3 exhibited higher chewiness, whereas C4 and C5 showed a reduction in chewiness. This observation can be correlated with the study of Dziki et al. (2016), who reported that incorporation of GCBp may cause significant alterations in WWB texture profiles. 

### 3.8. Total Phenolic Content (TPC)

The total phenolic content of the Control and GCBp-incorporated bread formulations was duly recorded. It was found that there was a positive correlation between the TPC content and GCBp content (Figure 9). C0 showed the lowest TPC value compared to all other formulations (140.47 ± 9.43 mg GAE/100 g, *p* < 0.05). C1 (172.50 ± 10.44 mg GAE/100 g), C2 (185.48 ± 8.94 mg GAE/100 g), and C3 exhibited a higher TPC content than C0. The values were statistically similar in C1, C2, and C3 (*p* > 0.05), which might correlate with the presence of other ingredients in the bread matrix, which may inhibit the interaction between the phenolic compounds and the bread matrix, resulting in similar phenolic content among these formulations. The findings can be corroborated with the observations reported by Dziki et al. (2015), who stated that there were similar results among those formulations that contained lower amounts of GCBp (1–3%) [57,72]. Furthermore, C4 (343.01 ± 11.62 mg GAE/100 g) and C5 (390.664 ± 14.38 mg GAE/100 g) displayed a corresponding increase in TPC value as the content of GCBp was gradually enhanced. Out of all the formulations, the TPC value of C5 was the highest (*p* < 0.05). One of the reasons for this phenomenon might be the modification in the internal structure of bread that promotes the housing of phenolic compounds by the bread matrix when GCBp is added at higher concentrations [73]. According to Masek et al. (2020), the phenolic compounds present in plant materials have various biological effects, especially high antioxidant and antibacterial potential [19]. Furthermore, previous reports state that chlorogenic and caffeic acid are the primary phenolic constituents in GCBp, and they possess anticancer, antimutagenic, antibacterial, and antioxidant attributes [20]. As a result, adding GCBp to bread formulations was associated with an increased TPC value, which may increase the antioxidant activity in the bread [22].

### 3.9. Microbiological Analysis

Microbiological study is a crucial parameter that implies the lifespan and overall quality of the food commodities [74]. In our study, the analysis was conducted on the 0th and 2nd days following the method carried out by Singh et al. (2024) [39]. The microbial analysis of bread formulations indicated that as the GCBp content increased, a corresponding decrease was observed in the total viable count (TVC). On the 0th day, it was noted that the TVC was highest in C0 (29.33 ± 3.21 × 10^3^ CFU/g) among all the formulations. In C1, TVC was reduced compared to C0 but was statistically similar (*p* > 0.05) (Figure 10C(i)). With a further increase in the GCBp content, C2 (16.00 ± 3.46 × 10^3^ CFU/g) and C3 (14.66 ± 3.16 × 10^3^ CFU/g) exhibited lower TVC compared to C0 and C1 (*p* < 0.05). With even higher GCBp content, C4 and C5 revealed a subsequent reduction in TVC compared to other formulations (*p* < 0.05). The TVC of C4 and C5 were similarly valued (*p* > 0.05).

Subsequently, on the 2nd day, the TVC of C0 was the highest (241.33 ± 5.84 × 10^3^ CFU/g, *p* < 0.05) among all the formulations (Figure 10C(ii)). Hasish et al. (2023) reported that the acceptable limit for TVC in baked goods like bread, biscuits, and cakes is less than 100 × 10^3^ CFU/g [75]. Our studies indicate that the TVC of C0 (without GCBp) exceeded the acceptable standards by the second day of storage, indicating spoilage of the bread. Conversely, it was noticed that increasing the GCBp content in bread caused a significant reduction in TVC in C1 (50.33 ± 7.57 × 10^3^ CFU/g), C2 (43.33 ± 2.08 × 10^3^ CFU/g), C3 (37.00 ± 5.19 × 10^3^ CFU/g), C4 (31.00 ± 2.00 × 10^3^ CFU/g), and C5 (24 ± 5.56 × 10^3^ CFU/g) compared to C0. The results showed that the TVC of all the GCBp-containing formulations remained below the maximum permitted limit, which can be correlated to the enhanced antimicrobial behavior of the GCBp-incorporated bread formulations, thereby suggesting that adding GCBp can prolong the shelf life of the WWB formulations. A previous investigation conducted by Ibrahim et al. (2020) stated that the chlorogenic acid of green coffee beans inhibits the growth of microorganisms in bread by disrupting their cell membranes and interfering with their metabolic process, which causes a reduction in the microbial growth during the process of storage, thereby extending durability and preserving overall bread attributes [21]. 

### 3.10. Sensory Evaluation

The results of sensory evaluation on a Hedonic scale of 1 to 5 on the WWB formulations are given in Table 5. Herein, the sensory evaluation was carried out by analyzing the appearance, aroma, and other parameters. It is reported that the average scores represent the sensory perception of the formulations exhibited by the consumers. Higher values reflect positive outcomes, and lower values indicate negative outcomes. It was noted that C1 (bread containing 1% GCBp) has an improved appearance compared to C0. Among the GCBp-containing formulations, C3, C4, and C5 reflected a lower score in appearance in comparison with the formulations C1 and C2 (*p* < 0.05). However, the values were similar in C3, C4, and C5 (*p* > 0.05). This suggests that the minimal concentration of GCBp had improved the appearance of C1, which could be attributed to the lighter color, improved porosity, and enhanced loaf volume found in the previous section of this study (Section 3.1 and Section 3.5). However, a high concentration of GCBp negatively impacted the appearance of C3, C4, and C5.

C1 exhibited the highest-rated aroma when compared with other formulations (*p* < 0.05). Incidentally, C0, C2, C3, C4, and C5 were similarly valued (*p* > 0.05). The MR between various volatile compounds of GCBp and sugar molecules leads to the characteristic aroma of bread [76]. This indicates that the minimal concentration of GCBp in bread had improved the aroma of C1. No significant variation was observed in visual palatability due to color appearance between C0, C1, and C2 (*p* > 0.05). Meanwhile, among GCBp-containing bread formulations, C3, C4, and C5 showed reduced visual palatability due to color appearance. This suggests that the addition of a low concentration of GCBp did not affect the visual palatability due to the color appearance of the bread. However, at high concentrations of GCBp, the visual palatability due to color appearance may be decreased. The softness of the bread did not show any significant difference among the formulations, which indicates that the addition of GCBp did not generate any change in the softness of the bread. 

Overall, the impression of C0, C1, and C2 was higher than of all other formulations. When the GCBp content was increased in C3, C4, and C5, a lower overall impression was observed. The overall impressions of C3, C4, and C5 were similarly valued (*p* > 0.05). Additionally, a five-point radar plot was created to improve the understanding of the sensory analysis through a diagrammatic representation (Figure 11). An observational analysis of the radar plot indicated similar sensory attributes of C1 to those of C0. On the other hand, among the GCBp-incorporated formulations, C1 showed the best attributes, while C5 showed the worst attributes.

## 4. Conclusions

This research focused on evaluating the impact of incorporating GCBp at different concentrations (1–5% *w*/*w*) on the physicochemical, biological, and sensory characteristics of WWBs. The comprehensive results from the study have revealed that the addition of GCBp influenced some vital quality parameters of the bread. C1 (1% GCBp), which contained the lowest GCBp content, exhibited the highest loaf volume. This suggests the optimal fermentation in the dough. On the other hand, a higher concentration of GCBp resulted in a decrease in loaf volume when compared with C1. Moisture analysis demonstrated a subtle difference in the moisture retention properties of the prepared formulations upon the incorporation of GCBp. C1 showed the highest impedance and a high degree of homogeneity among all the formulations, as observed from the (RQ)Q model parameters. Further, the addition of GCBp caused a notable reduction in the swelling percentage of formulations compared to Control (C0), which might be due to the inhibition of starch gelatinization by fatty acids and proteins in GCBp. The colorimetry study suggested that C1 displayed the highest L* and WI values, contributing to the least browning due to a lower proportion of GCBp content. In contrast, higher concentrations of GCBp led to increased browning in bread, possibly attributed to the MR. The FTIR analysis showed that C0, C1, and C2 exhibited improved intermolecular interactions within the bread matrix compared to other formulations. This may be due to the destabilization of the gluten matrix at higher concentrations of GCBp, which causes a reduction in the –OH stretching vibrations, thereby resulting in a decrement in the peak intensities corresponding to the –OH stretching vibrations. Additionally, except gumminess, most of the textural parameters, including hardness, springiness, cohesiveness, chewiness, and resilience, were not affected by the addition of GCBp. In the case of gumminess, C2 showed the highest value. The TPC levels of all the GCBp-incorporated formulations were higher than those of C0, which correlates with the improved antioxidant potential of the formulations. Furthermore, all the GCBp-containing formulations demonstrated an extended shelf life compared to the Control (C0). This could be due to the presence of chlorogenic acid, which has substantial antimicrobial activity, in GCBp. In the case of sensory analysis, it was found that C1 exhibited an optimal appearance and aroma compared to other formulations. Overall, C1 (containing 1% GCBp) exhibited the most beneficial physicochemical properties, such as the highest volume, improved color, enhanced intermolecular interactions, and better sensory attributes among all the formulations. Hence, it could be concluded that incorporating GCBp at a concentration of 1% appears to be optimal for promoting the quality of WWB while maintaining its sensory acceptability.

## Figures and Tables

**Figure 1 foods-13-02705-f001:**
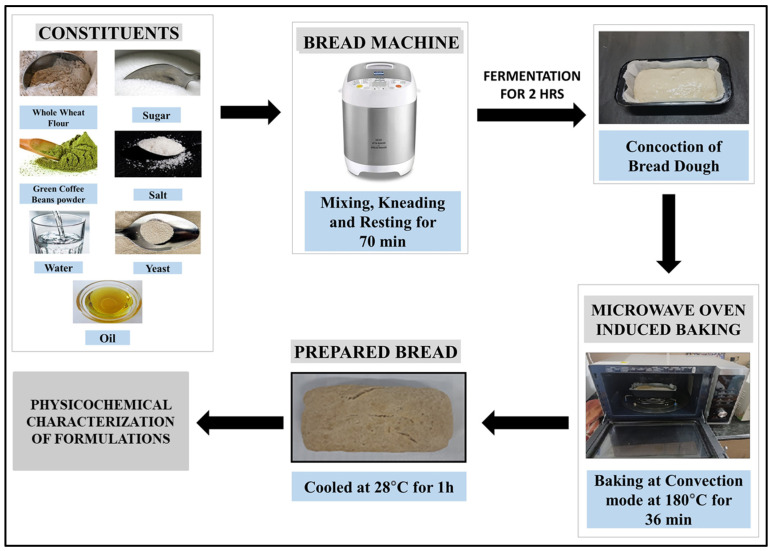
A flow diagram depicting the process of WWB formulations.

**Figure 2 foods-13-02705-f002:**
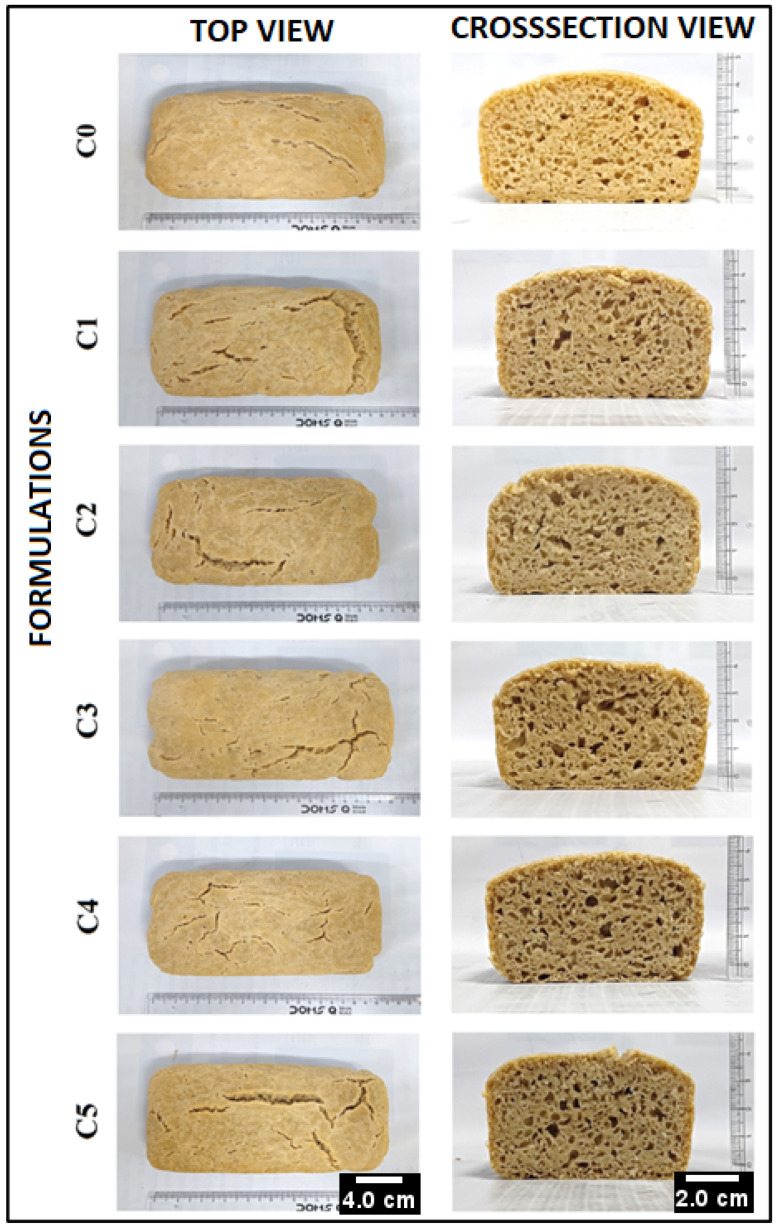
Visual appearance of the prepared bread formulations. C0 (Control, 0% GCBp), C1 (1% GCBp), C2 (2% GCBp), C3 (3% GCBp), C4 (4% GCBp), and C5 (5% GCBp).

**Figure 3 foods-13-02705-f003:**
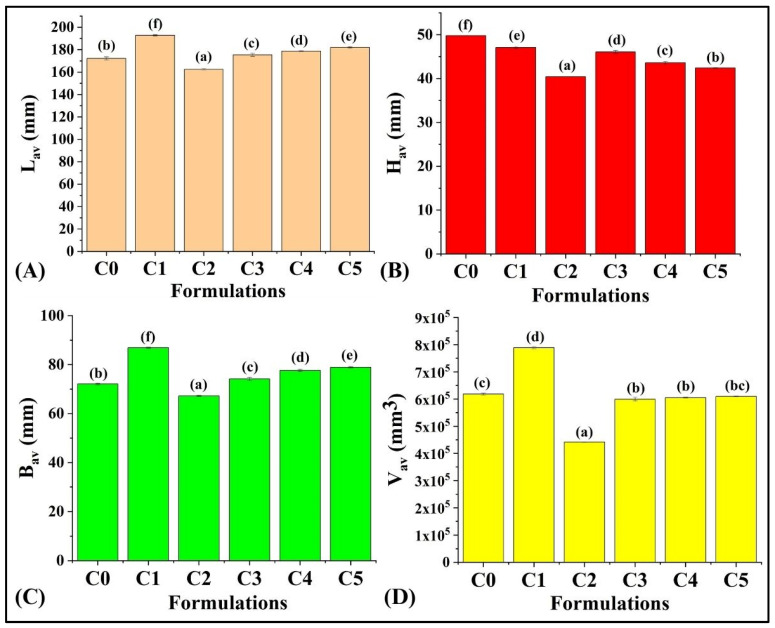
Physical dimensions of WWBs with different concentrations of GCBp. (**A**) L_av_, (**B**) B_av_, (**C**) H_av_, and (**D**) V_av_. Results are represented as average ± SD values of triplicate formulations. Respectively, statistical differences are shown via the bars assigned with different letters (*p* < 0.05).

**Figure 4 foods-13-02705-f004:**
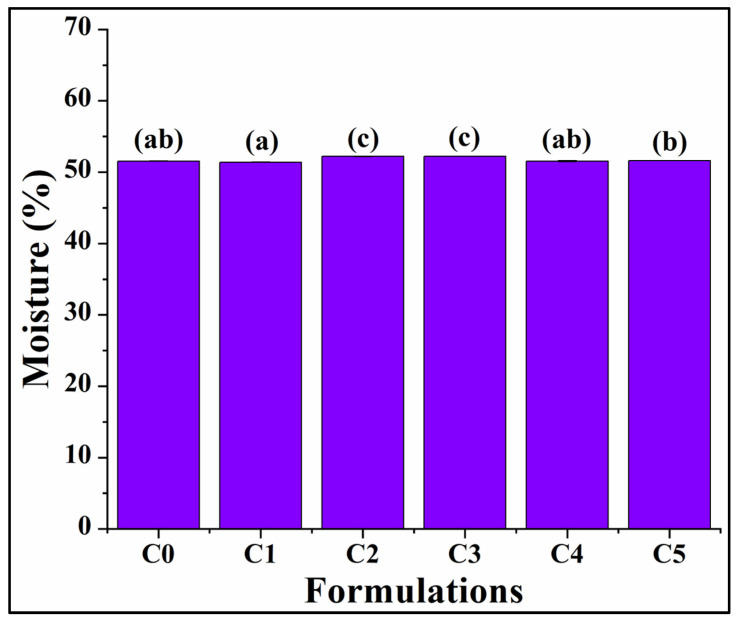
Graph representing the moisture content of the formulations. Results are represented as average ± SD values of triplicates. Respectively, bars assigned with different letters pinpoint significant differences (*p* < 0.05).

**Figure 5 foods-13-02705-f005:**
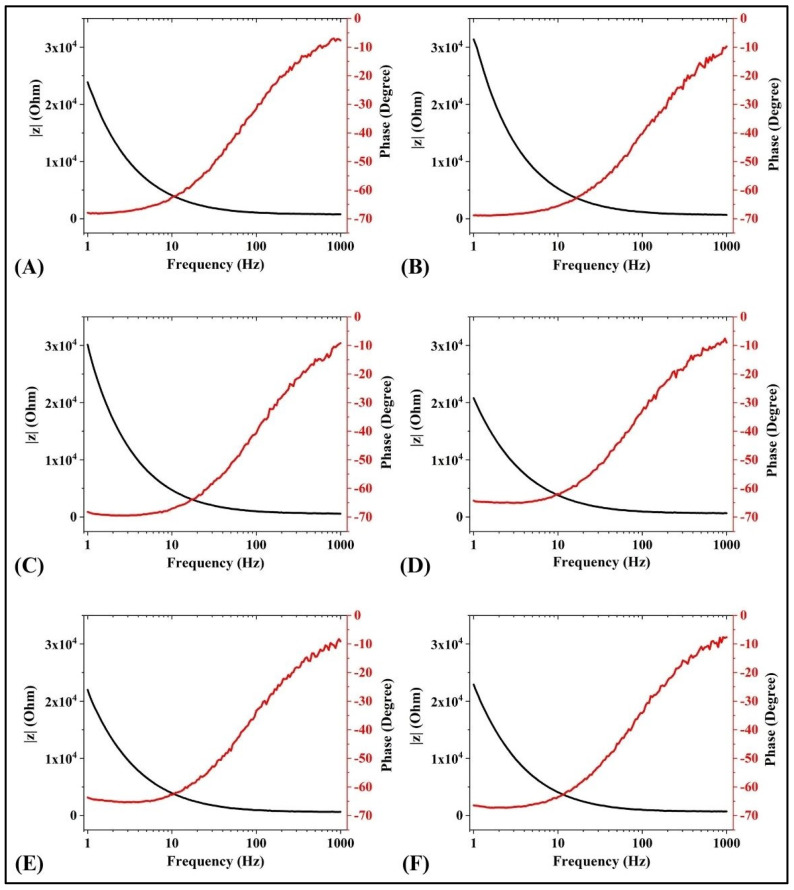
Bode plots of the impedance profiles of (**A**) C0, (**B**) C1, (**C**) C2, (**D**) C3, (**E**) C4, and (**F**) C5.

**Figure 6 foods-13-02705-f006:**
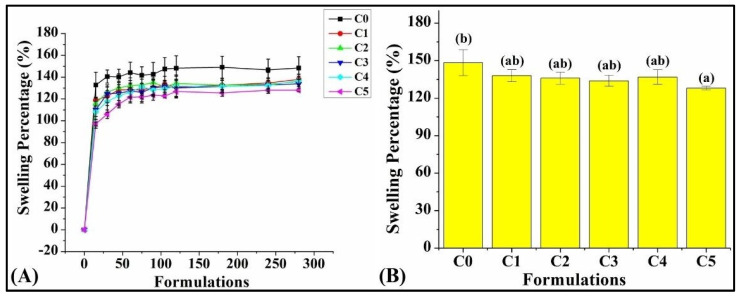
Graph showing (**A**) swelling percentage (%) profile, and (**B**) swelling percentage (%) after the culmination of the swelling study (5 h). The results are represented as average ± SD values of triplicates. Respectively, bars assigned with different letters pinpoint significant differences (*p* < 0.05).

**Figure 7 foods-13-02705-f007:**
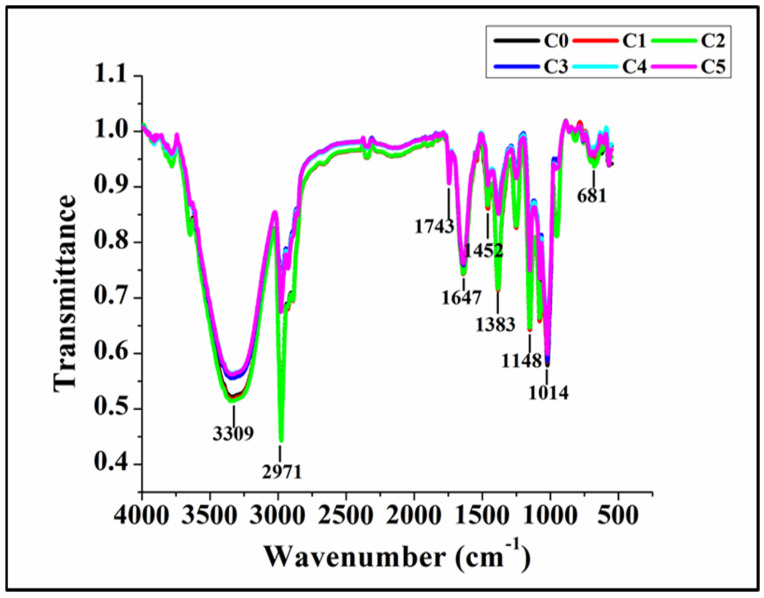
Graph representing FTIR spectral profiles of the formulations.

**Figure 8 foods-13-02705-f008:**
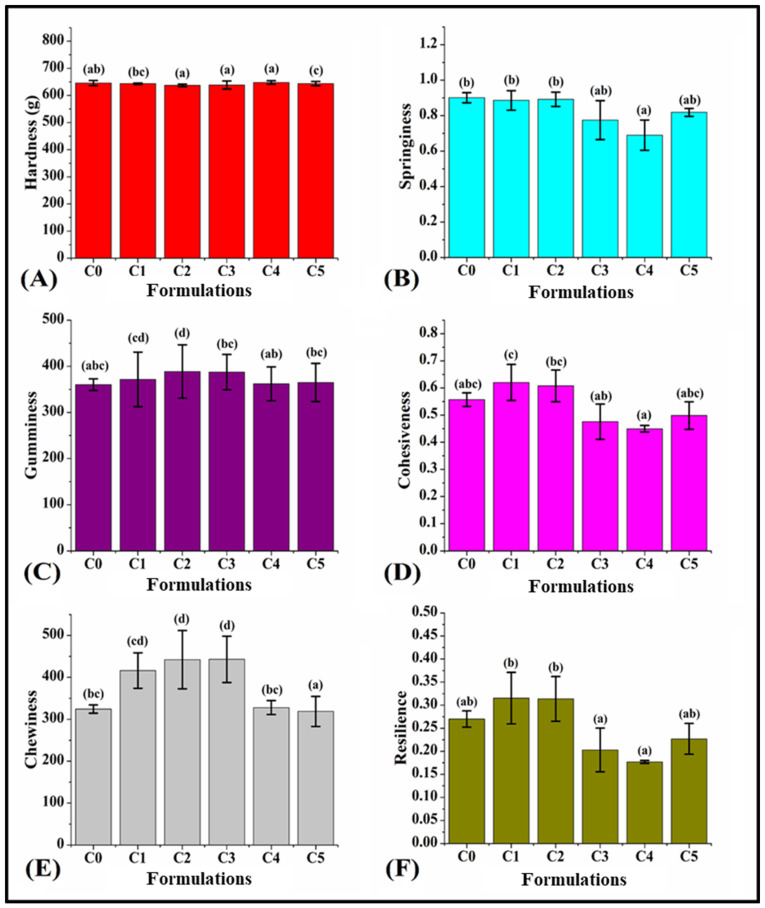
Graphs representing the TPA (texture profile analysis) parameters of the prepared formulations. (**A**) Hardness, (**B**) springiness, (**C**) gumminess, (**D**) cohesiveness, (**E**) chewiness, (**F**) resilience. The results are represented as average ± SD values of triplicate formulations. Respectively, bars assigned with different letters pinpoint significant differences (*p* < 0.05).

**Figure 9 foods-13-02705-f009:**
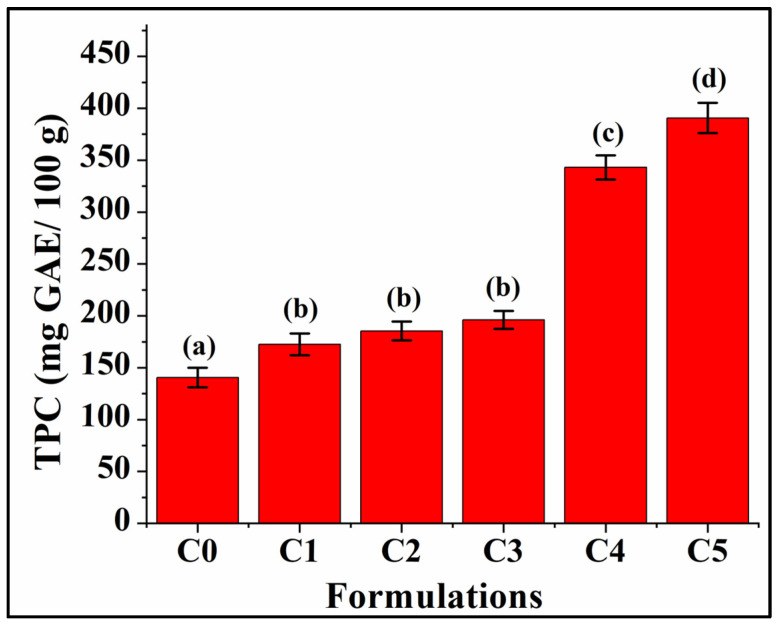
Graphs representing the total phenolic content of the formulations. Results are represented as average ± SD values of triplicates. Respectively, bars assigned with different letters pinpoint significant differences (*p* < 0.05).

**Figure 10 foods-13-02705-f010:**
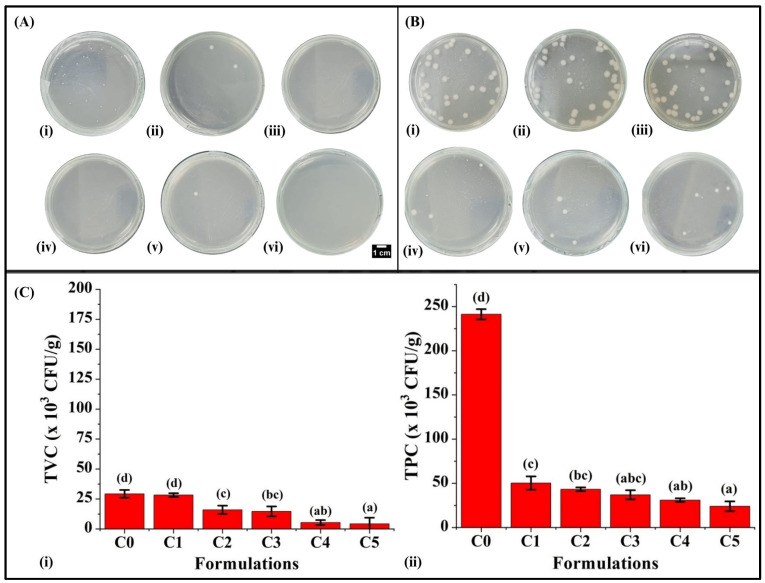
Results of the antimicrobial studies. (**A**) Colony count observed on the 0th day of storage: (**i**) C0, (**ii**) C1, (**iii**) C2, (**iv**) C3, (**v**) C4, and (**vi**) C5. (**B**) Colony count observed on 2nd day of storage: (**i**) C0, (**ii**) C1, (**iii**) C2, (**iv**) C3, (**v**) C4, and (**vi**) C5. (**C**) Graphs depicting the total viable count (×10^3^ CFU/g) of bread formulations: (**i**) on the 0th day of storage and (**ii**) on the 2nd day of storage. The results are represented as average ± SD values of triplicate formulations. Respectively, bars assigned with different letters pinpoint significant differences (*p* < 0.05).

**Figure 11 foods-13-02705-f011:**
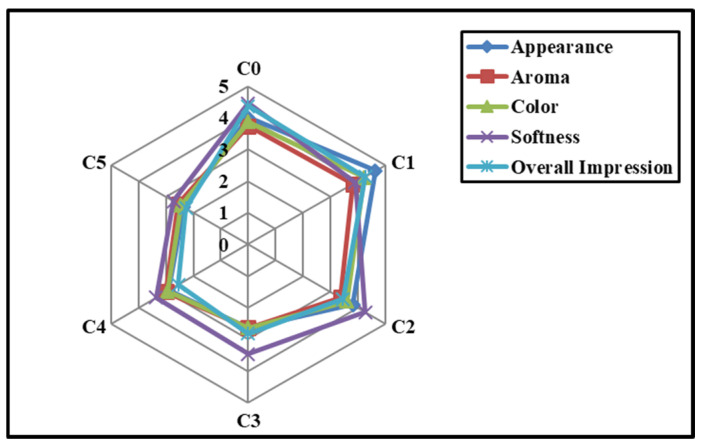
Radar plot representing the sensory attributes of WWB formulations on a five-point hedonic scale.

**Table 1 foods-13-02705-t001:** Composition of WWBs.

WWB Formulations	Composition						
	WWF (g)	GCBp (g)/(%)	Yeast (g)	Sugar (g)	Salt (g)	Oil (mL)	Water (mL)
C0	220.00	0.00 (0.00%)	7.00	29.00	2.00	22.00	220.00
C1	217.80	2.20 (1.00%)	7.00	29.00	2.00	22.00	220.00
C2	215.60	4.40 (2.00%)	7.00	29.00	2.00	22.00	220.00
C3	213.40	6.60 (3.00%)	7.00	29.00	2.00	22.00	220.00
C4	211.20	8.80 (4.00%)	7.00	29.00	2.00	22.00	220.00
C5	209.00	11.00 (5.00%)	7.00	29.00	2.00	22.00	220.00

**Table 2 foods-13-02705-t002:** Electrical parameters of the formulations.

Model Parameters	Formulations					
	C0	C1	C2	C3	C4	C5
**R (Ω)**	100,000 ± 0	100,000 ± 0	100,000 ± 0	100,000 ± 0	100,000 ± 0	100,000 ± 0
**Q (F)**	6.54 × 10^−5^ ± 1.49 × 10^−6 de^	1.16 × 10^−5^ ± 3.03 × 10^−6 a^	2.10 × 10^−5^ ± 1.00 × 10^−6 b^	3.20 × 10^−5^ ± 6.55 × 10^−6 c^	2.46 × 10^−5^ ± 4.08 × 10^−6 bc^	2.05 × 10^−5^ ± 6.84 × 10^−7 ab^
**Q1 (F)**	1.32 × 10^−5^ ± 6.88 × 10^−7 a^	2.72 × 10^−5^ ± 2.15 × 10^−5 a^	1.17 × 10^−5^ ± 5.77 × 10^−7 a^	1.33 × 10^−5^ ± 2.34 × 10^−6a^	1.46 × 10^−5^ ± 1.77 × 10^−6 a^	1.49 × 10^−5^ ± 9.52 × 10^−7 a^
**n**	0.7500 ± 0.0500 ^bc^	0.7633 ± 0.0764 ^c^	0.7567 ± 0.0153 ^c^	0.5867 ± 0.0115 ^a^	0.6267 ± 0.0569 ^ab^	0.6533 ± 0.0153 ^abc^
**n_1_**	0.7166 ± 0.0058 ^a^	0.7860 ± 0.0909 ^a^	0.8190 ± 0.0115 ^a^	0.9908 ± 0.0160 ^b^	1.0000 ± 0.0000 ^b^	1.0000 ± 0.0000 ^b^
**R^2^**	0.9988 ± 0.0007 ^a^	0.9994 ± 0.0005 ^a^	0.9998 ± 0.0000 ^a^	0.9997 ± 0.0002 ^a^	0.9996 ± 0.0003 ^a^	0.9997 ± 0.0002 ^a^

NB: 1. Different letters signify statistical significance (*p* < 0.05) in the row. 2. (a) R—resistance linked with the overall characteristics of the formulations, (b) Q—constant phase element linked with the overall attributes of formulations, (c) Q1—constant phase element linked with the overall attributes of the formulation electrode junctions, (d) n—homogeneity constant linked with Q, (e) n1—homogeneity constant linked with Q1.

**Table 3 foods-13-02705-t003:** Color parameters obtained through colorimetric studies.

WWB Formulations	Color Parameters					
	L*	a*	b*	WI	YI	BI
C0	61.73 ± 1.07 ^b^	6.55 ± 1.61 ^a^	44.91 ± 1.14 ^a^	40.60 ± 1.29 ^b^	103.98 ± 3.75 ^b^	124.56 ± 7.10 ^b^
C1	72.34 ± 1.4 ^c^	6.39 ± 1.01 ^a^	41.68 ± 1.18 ^a^	49.53 ± 0.31 ^c^	82.31 ± 0.86 ^a^	88.15 ± 1.18 ^a^
C2	65.21 ± 0.21 ^b^	6.57 ± 0.47 ^a^	44.97 ± 3.40 ^a^	40.86 ± 3.10 ^b^	103.37 ± 9.20 ^b^	123.89 ± 15.88 ^b^
C3	67.31 ± 0.36 ^b^	7.90 ± 0.40 ^ab^	59.66 ± 2.57 ^b^	28.62 ± 2.38 ^a^	138.30 ± 6.84 ^c^	204.98 ± 19.77 ^c^
C4	59.77±0.83 ^ab^	10.52 ±1.20 ^b^	59.91 ± 1.20 ^b^	27.05 ± 1.22 ^a^	143.23 ± 4.33 ^c^	222.38 ± 12.47 ^c^
C5	58.11 ± 0.83 ^a^	6.19 ± 1.20 ^a^	52.98 ± 1.03 ^b^	26.28 ± 0.41 ^a^	148.29 ± 1.17 ^c^	233.70 ± 2.58 ^c^

NB: Different letters signify statistical significance (*p* < 0.05) in the row.

**Table 4 foods-13-02705-t004:** Color parameters obtained through microcolorimetric studies.

WWB Formulations	Color Parameters					
	L_m_*	a_m_*	b_m_*	WI_m_	YI_m_	BI_m_
C0	73.36 ± 0.70 ^cd^	8.09 ± 0.94 ^a^	14.82 ± 2.15 ^ab^	59.64 ± 1.96 ^bc^	52.33 ± 6.71 ^ab^	55.09 ± 7.01 ^ab^
C1	77.97 ± 2.47 ^d^	8.60 ± 0.17 ^a^	11.19 ± 1.72 ^a^	70.73 ± 1.71 ^c^	18.73 ± 9.07 ^a^	22.74 ± 7.06 ^a^
C2	72.09 ± 2.27 ^bc^	8.88 ± 1.07 ^a^	16.85 ± 1.10 ^bc^	67.32 ± 3.17 ^b^	26.12 ± 11.46 ^bc^	29.21 ± 10.56 ^bc^
C3	67.80 ± 0.49 ^abc^	9.56 ± 0.82 ^ab^	15.15 ± 2.71 ^ab^	71.44 ± 0.87 ^ab^	16.38 ± 6.86 ^b^	21.37 ± 5.46 ^bc^
C4	67.01 ± 0.60 ^ab^	13.87 ± 3.73 ^b^	18.73 ± 1.22 ^bc^	78.11 ± 1.83 ^a^	13.48 ± 5.61 ^bc^	15.49 ± 4.25 ^d^
C5	65.38 ± 3.55 ^a^	7.93 ± 0.21 ^b^	20.27 ± 0.90 ^c^	68.46 ± 3.16 ^a^	24.43 ± 11.91 ^c^	26.07 ± 11.50 ^cd^

NB: Different letters signify statistical significance (*p* < 0.05) in the row.

**Table 5 foods-13-02705-t005:** Average scores obtained from the sensory evaluation of the formulations.

Sensory Attributes	Formulations					
	C0	C1	C2	C3	C4	C5
**Appearance**	4.00 ± 0.63 ^bc^	4.63 ± 0.67 ^c^	3.81 ± 0.63 ^bc^	2.63 ± 0.80 ^a^	2.90 ± 1.10 ^ab^	2.36 ± 1.36 ^a^
**Aroma**	3.72 ± 1.10 ^a^	3.81 ± 0.87 ^b^	3.36 ± 1.28 ^a^	2.63 ± 0.80 ^a^	3.00 ± 1.18 ^a^	2.54 ± 0.82 ^a^
**Color**	4.54 ± 0.68 ^b^	4.36 ± 0.67 ^b^	4.00 ± 0.63 ^b^	2.81 ± 0.80 ^a^	2.72 ± 1.00 ^a^	2.09 ± 1.13 ^a^
**Softness**	4.45 ± 0.68 ^b^	3.90 ± 0.70 ^ab^	4.27 ± 1.00 ^b^	3.50 ± 1.21 ^ab^	3.36 ± 1.20 ^ab^	2.72 ± 1.34 ^a^
**Overall Impression**	4.36 ± 0.80 ^b^	4.27 ± 0.64 ^b^	3.45 ± 1.12 ^ab^	2.81 ± 0.87 ^a^	2.54 ± 0.03 ^a^	2.27 ± 1.00 ^a^

NB: Different letters signify statistical significance (*p* < 0.05) in the row.

## Data Availability

The original contributions presented in the study are included in the article, further inquiries can be directed to the corresponding author.

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
