# Peer review of "Evaluating the Impact of Green Coffee Bean Powder on the Quality of Whole Wheat Bread: A Comprehensive Analysis"

_foods, 2024, doi:10.3390/foods13172705_

Round 1
Reviewer 1 Report
Comments and Suggestions for Authors
The paper is well done and described, it is easy to understand as well and is appropriate for the journal. The authors did a good job.
Line 23: did the authors a statistical analysis? because is they did I recommended not express the term “similar”.
Key words: I suggest add the word bread
Line 57: Is it correct to express tolerable dietary fiber? If is not, change the redaction
Line 97-102: I suggest to delete the lines, these are more like a discussion that an introduction
Table 1: the authors have to check the format for tables and figures and make corrections if is necessary
Line 215: delete word duration
Line 216:
Be specific about the temperature, because ambient temperature is to general
Describe the equipment and temperature of the centrifugation
Line 225: use a synonymous of thoroughly
Line 236: extra space after 3 times
Line 309: the term statistically similarity to (p > 0.05) have to change to no significant difference were showed or similar, is not correct to express statistically similarity. Check in the result section.
Line 422: the section Colorimetry I suggest to express the results f L, a* y b* in table not in a graphic.

Author Response
Reviewer-1
The paper is well done and described, it is easy to understand as well and is appropriate for the journal. The authors did a good job.
Comment 1: Line 23: did the authors a statistical analysis? because is they did I recommended not express the term “similar”.
Response 1: As per the suggestion of the learned reviewer, we have made the correction accordingly in Lines 22 and 23.
Comment 2: Key words: I suggest add the word bread
Response 2: As per the suggestion of the learned reviewer, we have added the word in the Keywords section (L35).
Comment 3: Line 57: Is it correct to express tolerable dietary fiber? If is not, change the redaction
Response 3: As per the suggestion of the learned reviewer, we have removed the term and made the correction accordingly in Line 59.
Comment 4: Line 97-102: I suggest to delete the lines, these are more like a discussion that an introduction
Response 4: As per the suggestion of the learned reviewer, we have shortened the introduction section and made the corrections accordingly.
Comment 5: Table 1: the authors have to check the format for tables and figures and make corrections if is necessary
Response 5: As per the suggestion of the learned reviewer, we have made corrections to all the tables and figures in the requisite format, according to the journal guidelines.
Comment 6: Line 215: delete word duration
Response 6: As per the suggestion of the learned reviewer, we have made the correction accordingly in Line 224.
Comment 7: Line 216:
Be specific about the temperature, because ambient temperature is to general
Describe the equipment and temperature of the centrifugation
Response 7: As per the suggestion of the learned reviewer, we have mentioned the temperature in Line 224. We have also added the model, manufacturer, and country of the equipment and the temperature used in centrifugation in lines 225 and 226, respectively.
Comment 8: Line 225: use a synonymous of thoroughly
Response 8: As per the suggestion of the learned reviewer, we have made the correction accordingly in Line 235.
Comment 9: Line 236: extra space after 3 times
Response 9: As per the suggestion of the learned reviewer, we have made the correction accordingly in Line 246.
Comment 10: Line 309: the term statistically similarity to (p > 0.05) have to change to no significant difference were showed or similar, is not correct to express statistically similarity. Check in the result section.
Response 10: We apologize for the mistake. As per the suggestion of the learned reviewer, we have made the corrections accordingly in Lines 340-351. (There is no term of statistically similarity here as the results have changed after complying with the reviewer 2 comments).
Comment 11: Line 422: the section Colorimetry I suggest to express the results f L, a* y b* in table not in a graphic.
Response 11: As per the suggestion of the learned reviewer, we have removed the graphs from section 3.5. Colorimetry and Microcolorimetry represented the results of color parameters in a tabular format (Table 3 and Table 4), following the journal guidelines.

Reviewer 2 Report
Comments and Suggestions for Authors
Line 292. Physical examination is very subjective because depends of the initial weight of the sample, time, etc. To do some discussion about this measurement is not possible because is only one sample. My ask to author is that make the measurements to three different bread samples (replication) and that not make three repetitions with the Vernier on the same bread sample. With these measurements, DS and the effect of the green coffee powder will have significance difference.
Line 389. Figure4 b show the module of the impedance measurement (|z|). Normally, impedance measurement is not a value, really is a vector with module Z and angle. Sometimes authors report the value of resistance and capacitance obtained from impedance measurements to show the effect of the air bubbles. Other authors do EIT and from those measurements (Bode diagrams) report the equivalent circuit which is associated to volume. In this case author measure module Z and say that module Z is the highest for C1 because C1 has the highest volume. Then author must apply EIT to find the equivalent circuit and to find the relation with the volume. So, explain in text the relation between impedance and volume.
Line 500. FTIR show that bread samples are divided in two groups: One group with the same spectra (C00, C1 y C2) and other (C3,C4,C5). I do not know why. I believe that is an effect in the baking process. Discuss in text.
Line 689. Author must mention that sensory evaluation was approved by an ethic committee.
Author Response
Reviewer-2
Comment 1: Line 292. Physical examination is very subjective because depends of the initial weight of the sample, time, etc. To do some discussion about this measurement is not possible because is only one sample. My ask to author is that make the measurements to three different bread samples (replication) and that not make three repetitions with the Vernier on the same bread sample. With these measurements, DS and the effect of the green coffee powder will have significance difference.
Response 1: We appreciate your comment and apologize for being unable to extend our thoughts. As per the suggestions of the learned reviewer, the experiments were conducted, and the physical dimensions were recorded in triplicate samples. It was observed that there were significant differences among the samples, although, as reported previously, C1 retained the highest volume among all the samples. We have incorporated the corrections accordingly in section 3.1 in Lines 302-351.
Comment 2: Line 389. Figure4 b show the module of the impedance measurement (|z|). Normally, impedance measurement is not a value, really is a vector with module Z and angle. Sometimes authors report the value of resistance and capacitance obtained from impedance measurements to show the effect of the air bubbles. Other authors do EIT and from those measurements (Bode diagrams) report the equivalent circuit which is associated to volume. In this case author measure module Z and say that module Z is the highest for C1 because C1 has the highest volume. Then author must apply EIT to find the equivalent circuit and to find the relation with the volume. So, explain in text the relation between impedance and volume.
Response 2: As per the suggestions of the learned reviewer, the Bode plots representing the impedance and phase responses were plotted, and subsequently, we applied the (RQ)Q model to find out several electrical parameters to explain the relation between impedance and volume (Lines 400-427).
Comment 3: Line 500. FTIR show that bread samples are divided in two groups: One group with the same spectra (C00, C1 y C2) and other (C3, C4, C5). I do not know why. I believe that is an effect in the baking process. Discuss in text.
Response 3: As per the suggestion of the learned reviewer, we have discussed the reason behind this phenomenon and made the corrections accordingly in Lines 548-553.
Comment 4: Line 689. Author must mention that sensory evaluation was approved by an ethic committee.
Response 4: As per the suggestion of the learned reviewer, we have incorporated the approval of the institutional ethical committee for conducting sensory analysis (Lines 814-817).

Reviewer 3 Report
Comments and Suggestions for Authors
Through comprehensive measurements, the authors showed that the addition of coffee bean flour can improve the functionality and shelf life of breads, while maintaining their quality. The results of this study will be of interest to many people, including bakers and health-conscious consumers. I will offer only a few comments on the content of this study.
Comment 1.
Please provide a more detailed description of the volume calculations. According to Figure 2, the shape of the bread does not appear to be an exact rectangle. Did you multiply the length, width, and width by the length of the pan to obtain the volume or did you perform a correction calculation?
Comment 2.
In lines 341, 343, and 345, “mm3” is listed and the “3” is not superscripted. Please correct them.
Author Response
Reviewer-3
Through comprehensive measurements, the authors showed that the addition of coffee bean flour can improve the functionality and shelf life of breads, while maintaining their quality. The results of this study will be of interest to many people, including bakers and health-conscious consumers. I will offer only a few comments on the content of this study.
Comment 1: Please provide a more detailed description of the volume calculations. According to Figure 2, the shape of the bread does not appear to be an exact rectangle. Did you multiply the length, width, and width by the length of the pan to obtain the volume or did you perform a correction calculation?
Response 1: We appreciate your comment and apologize for being unable to extend our thoughts. As per the suggestion of the learned reviewer, we have introduced section 2.2.2, wherein we have described the methodology for calculating the volume from the length, breadth, and height of the prepared samples mentioned in Lines 138-149.
Comment 2: In lines 341, 343, and 345, "mm3" is listed and the "3" is not superscripted. Please correct them.
Response 2: As per the suggestion of the learned reviewer, we have made the corrections accordingly in Lines 342, 344, and 346.

Round 2
Reviewer 2 Report
Comments and Suggestions for Authors
Title. Title must change because in this article there were not any optimizing wheat bread quality. Author made different concentration of GCB and never optimized any parameter.